# Borosilicate Glass-Ceramics Containing Zirconolite and Powellite for RE- and Mo-Rich Nuclear Waste Immobilization

**DOI:** 10.3390/ma14195747

**Published:** 2021-10-01

**Authors:** Wei Wan, Yongchang Zhu, Xingquan Zhang, Debo Yang, Yonglin Huo, Chong Xu, Hongfu Yu, Jian Zhao, Jichuan Huo, Baojian Meng

**Affiliations:** 1Fundamental Science on Nuclear Wastes and Environmental Safety Laboratory, Southwest University of Science and Technology, Mianyang 621010, China; wanwei992695714@163.com (W.W.); hyl20201016@163.com (Y.H.); yjcbma@163.com (C.X.); yuhongfu2018@mails.swust.edu.cn (H.Y.); z18210433040@163.com (J.Z.); huojichuan@swust.edu.cn (J.H.); 2China Building Materials Academy, Beijing 100024, China; yangdebo19770904@126.com (D.Y.); Baojian_meng@163.com (B.M.); 3School of Materials Science and Engineering, Southwest University of Science and Technology, Mianyang 621010, China

**Keywords:** multi-phase glass-ceramics, in situ heat treatment, zirconolite, powellite

## Abstract

In order to increase the loading of rare earth- and molybdenum-rich high-level waste in the waste forms, zirconolite- and powellite-based multi-phase borosilicate glass-ceramics were synthesized via an in-situ heat treatment method. The effects of the CTZ (CaO, TiO_2_ and ZrO_2_) content on the crystallization, microstructure and aqueous durability of the multi-phase borosilicate glass-ceramics were studied. The results indicate that the increase of CTZ content can promote crystallization. The glass-ceramics presented even structures when the CTZ content was ≥ 40 wt%. For the glass-ceramic with 40 wt% CTZ, only zirconolite and powellite crystals were detected and powellite crystals were mainly distributed around zirconolite, whereas for the glass-ceramics with 50 wt% CTZ, perovskite was detected. Furthermore, the leaching rates of Na, Ca, Mo and Nd were in the ×10^−3^, ×10^−4^, ×10^−3^ and ×10^−5^ g·m^−2^·d·^−1^ orders of magnitude on the 28th leaching day, respectively.

## 1. Introduction

Nuclear energy as a net-zero carbon emission energy source brings convenience to industry and life, but a large amount of spent fuel is produced in the process of producing nuclear energy. Different countries have different ways of disposing of spent fuel. Some countries, including the US, Canada and Sweden, have decided to leave their spent fuel in interim storage, whereas other countries such as France, Russia, the UK and Japan reprocess their spent nuclear fuel in order to recycle U and Pu that is used in new nuclear fuel and to reduce the radiotoxicity of the ultimate waste [1,2,3]. China has also decided to reprocess spent fuel. The reprocessing of spent fuel will produce a large quantity of high-level waste (HLW) containing rare earths (RE), minor actinides, transition metals, alkalis, alkaline earths and other fission products [4,5]. Most importantly, there are many radionuclides in HLW, which have strong radiotoxicity and can cause heritable damage to the organism. Once these radionuclides spill, they will have a devastating impact on the biosphere. Currently, how to immobilize HLW safely is a hot topic for researchers. 

In order to immobilize the waste streams safely, researchers have developed a series of waste forms such as alleged crystal-free glass, ceramics and glass-ceramics [6]. Vitrification is the most widely used HLW immobilization disposal method around the world [7] and has been successfully commercialized. At present, the solubility of HLW in glass is about 15 wt%–20 wt% [1]. However, the low solubility of MoO_3_ in borosilicate glass [8,9,10,11,12,13,14,15,16], due to its high field strength, limits the high-level radioactive waste loading in borosilicate glass [17]. These molybdates may precipitate out in the process of the melt cooling or heat treatment of glass. Although Mo is a kind of non-radioactive fission product, it tends to form molybdates with poor durability such as Na_2_MoO_4_ and ^137^Cs_2_MoO_4_ (known as yellow phase). A large number of studies about molybdate have been reported [15,18,19], but there is still no significant increase in HLW loading in glass waste forms. For these reasons, glass-ceramics have been developed and are regarded as potential materials to immobilize fission products so as to increase the HLW loading.

Research has been conducted to increase the HLW loading. For example, the US Department of Energy (DOE) developed a new glass-ceramic waste form containing oxyapatite, powellite and rare earth borosilicate crystals to immobilize the fission products. These glass-ceramics with stable crystalline phases are deemed to be the ideal materials for RE- and Mo-rich HLW immobilization [6,20,21]. Based on this knowledge, much research has been conducted. McCloy reported the effects of different components on the glass structure and crystallization [6]. Crum reported the effects of in situ heat treatment at different temperatures on the crystallization of glass-ceramics [1]. Neeway et al. evaluated the dissolution kinetics of oxyapatite and powellite at different pH and the chemical stability of glass-ceramics prepared at different cooling rates [22,23]. Even the influence of the size of rare-earth ions on the phase evolution of Mo-containing glass-ceramics has been considered [13]. The loading of Nd_2_O_3_ in glass-ceramics containing oxyapatite is apparently higher than that of glass waste forms. However, McCloy [6] and Patil [13] also reported uncontrolled crystallization and surface crystallization of oxyapatite, which may cause uncertainty in the leaching behavior. Moreover, the anti-irradiation property of oxyapatite is not particularly suitable [24]. Therefore, developing new glass-ceramic waste forms to immobilize HLW is imperative. 

Zirconolite (CaZrTi_2_O_7_) as a stable mineral found in nature has been extensively studied by scientists [25,26,27]. Due to its good chemical stability and thermal and irradiation properties, zirconolite is regarded as the crystalline phase with the most potential for HLW immobilization [28]. Zirconolite can be synthesized by adding relatively high amounts of TiO_2_ and ZrO_2_ into the glass as nucleating agents and then combining them with the corresponding heat treatment. Previous studies revealed that RE cations can replace both Ca^2+^ and Zr^4+^ sites of zirconolite structure [2,29], which enables RE to exist in the lattice of zirconolite, forming a stable crystalline phase. In addition, Loiseau reported that Nd^3+^ and Gd^3+^ can replace more than 60% of the Ca^2+^ sites of zirconolite without any structure distortion [30]. Therefore, zirconolite is considered a suitable material for Nd^3+^ immobilization. With zirconolite as a target crystal used for the immobilization of Nd_2_O_3_, the optimal content of CTZ should be ascertained because an insufficient amount of zirconolite cannot provide enough sites for Nd, or superfluous CTZ content causes uncontrollable crystals and more unnecessary grain boundaries. Early zirconolite glass-ceramics were synthesized by the two-step method [30,31,32], which enabled glass melts to cool and then be reheated to nucleation and crystallization temperature. However, this process is not suitable for engineering because of its complex operation process. According to the DOE plan, the best way to achieve HLW immobilization is to accomplish the technological process in canisters, which allows the target crystals to form in one step. In order to maximally simplify the heat treatment process, this study was designed to synthesis the glass-ceramics containing zirconolite and powellite by in situ thermal treatment. 

In consideration of the high content of RE and Mo, developing multi-phase glass-ceramic waste forms containing zirconolite and powellite is extremely urgent. Zirconolite for RE and powellite for Mo both exhibit good chemical durability. For example, the normalized leaching rates of zirconolite and powellite are generally below ×10^−4^ and ×10^−2^ g·m^−2^·d·^−1^ orders of magnitude after 28 d, respectively [22,33]. Glass-ceramic waste forms used in engineering should have good microstructures and uniformity. For example, the grain size is within the acceptable range and evenly distributed, which is strongly related to composition and heat treatment. In order to maximally simplify the preparation process of glass-ceramics, in situ heat treatment was performed. Therefore, the key is the composition of the glass-ceramics, especially the content of CTZ. 

In this study, according to the 35 wt% HLW loading required by the national defense program, RE and Mo are considered key elements. All the RE included in HLW are replaced by Nd since their particle radiuses and chemical properties are similar. In order to study the effects of CTZ content on the crystalline phases, microstructure and aqueous durability of multi-phase borosilicate glass-ceramics containing zirconolite and powellite, a series of multi-phase borosilicate glass-ceramics containing zirconolite and powellite with different CTZ content were synthesized to immobilize the RE and Mo of HLW generated from a nuclear power plant in China by in situ heat treatment. This paper aims to (1) understand the relationship between CTZ content and the crystalline phase, microstructure and bulk density of the multi-phase borosilicate glass-ceramics, (2) determine the composition of the multi-phase glass-ceramic waste forms with optimum CTZ content and (3) evaluate the durability of the multi-phase glass-ceramics.

## 2. Experimental Procedures

Glass-ceramics containing different CTZ content were synthesized using a reagent grade of SiO_2_, H_3_BO_3_, Na_2_CO_3_, Al(OH)_3_, CaCO_3_, TiO_2_, ZrSiO_4_, Nd_2_O_3_ and MoO_3_ as raw materials. In order to study the effects of CTZ content on the crystallization, microstructure and chemical stability of multi-phase glass-ceramics, different levels of CTZ content (20, 30, 40 and 50 wt%) were added. It should be noted that CaCO_3_ was designed to be divided into two parts, one for participating in the glass network and the formation of powellite crystals, the other part participating in the formation of zirconolite crystals. Table 1 shows the detailed compositions. All the ingredients were mixed and then held at 850 °C for 2 h to release carbon dioxide, before being melted in alumina crucibles at the temperature of 1250 °C for 3 h. All glass melts were poured into water, dried, ground into powder and reheated to 1250 °C for 2 h before in situ thermal treatment (slow cooling rate at about 5 °C·min^−1^). The process of glass-ceramics synthesis is shown in Figure 1. The main ingredients of HLW are RE (27.02 wt%) and Mo (13.28 wt%), which account for a large proportion of the total mass. Thus, the key point of HLW immobilization can be simplified as the immobilization of Nd and Mo.

In order to confirm the glass transition temperature (*T*_g_) as well as crystallization temperature (*T*_c_), the quenched glass samples were tested by differential thermal analysis (DTA, SDT Q600, TA Instruments Inc., New Castle, DE, USA) with a heating rate of 20 °C·min^−1^ in the air. All these samples were measured from room temperature to 1000 °C. The amorphous and crystalline phases of the glass-ceramics were identified by X-ray Diffraction (XRD, PANalytical X’Pert Pro diffractometer, Eindhoven, Holland) using Cu Kα radiation, operating at 40 kV and 40 mA, ranging from 3° to 80°. The microstructure and element distribution of the glass-ceramics were identified using a scanning electron microscope (SEM, Ultra-55, ZEISS Company, Jena, Germany) and energy-dispersive X-ray spectroscopy (EDS, IE450X-Max80, Oxford Instruments Analytical Limited, Oxford, UK). The densities of the samples were measured by the pycnometer method. All samples were dried in an 80 °C oven for 2 h until the mass did not change. The following formula (1) is used to calculate the density of the sample and repeated 3 times to take the average (g·cm^−3^):(1)$$\rho_{0}=\frac{m_{0}}{m_{0}+m_{1}-m_{2}}\times \rho_{1}$$
where *m*_0_ is the mass of the dry sample to be measured (g), *m*_1_ is the mass of the pycnometer filled with deionized water (g), *m*_2_ is the mass of the pycnometer filled with deionized water and the added sample (g) and *ρ*_1_ is the density of water that is decided by its temperature (g·cm^−3^).

The chemical durability of the obtained glass-ceramic samples was investigated at 90 ± 1.0 °C in deionized water (pH = 7) in PTFE reactors, according to the Product Consistency Test (PCT) [34]. Glass-ceramics were ground into powder and passed through a 100–200 mesh sieve, cleaned with absolute ethanol and dried. Then, a 3 g sample was soaked in 80 mL of deionized water in a PTFE reactor at 90 ± 1.0 °C [29]. After 1, 3, 7, 14 and 28 days, the leachates were removed for measurement and 80 mL of fresh deionized water was added to the PTFE. Inductively coupled plasma-optical emission spectroscopy (ICP-OES, iCAP6500, Thermo Fisher Company, Waltham, MA, USA) and inductively coupled plasma-mass spectrometry (ICP-MS, 7700X, Agilent Technologies, Santa Clara, CA, USA) were performed to test the obtained leachate. The normalized leaching rate *LR*_i_ (g·m^−2^·d^−1^) was calculated according to the formula (2):(2)$$LR_{i}=\frac{C_{i}\cdot V}{f_{i}\cdot S\cdot △t}$$
where *C_i_* is the concentration of element *i* in the leachate (g·L^−1^), *f_i_* is the mass fraction of the element *i* in the sample, △*t* is the interval leaching duration in days (d), *V* is the volume of the leachate (L) and *S* is the surface area of the sample (m^2^).

## 3. Results and Discussion

Figure 2 shows the DTA curves of the glass with different CTZ content. The *T*_g_ values of CTZ-20, CTZ-30, CTZ-40 and CTZ-50 are 605 °C, 654 °C, 661 °C and 667 °C, respectively. The result is similar to Li’s [32]. For the CTZ-20 sample, there are two weak exothermic peaks at around 725 °C and 940 °C. The peak at around 940 °C corresponds to the crystallization of powellite [16]. In addition, the DTA curves of CTZ-30, CTZ-40 and CTZ-50 are similar. All of the above have two exothermic peaks at around 776 °C and 941 °C, which corresponds to the crystallization of zirconolite [17] and powellite phase, respectively, and the peak at 776 °C strengthened with the increasing content of CTZ. Additionally, the peak at 725 °C disappeared when the CTZ content increased to 30 wt%–50 wt%, which demonstrated that 20 wt% CTZ content cannot meet the requirements of zirconolite crystallization, and another crystalline phase formed. The results indicate that an increase in CTZ content is beneficial to the crystallization of multi-phase glass-ceramics.

Figure 3 presents the XRD patterns of the glass-ceramics. For the CTZ-20 sample, it can be seen that the main crystalline phase is neodymium aluminum perovskite (NdAlO_3_, PDF#76-1336) [35], and the weak characteristic peaks belonging to powellite (CaMoO_4_, PDF#85-0585) and zirconolite (CaZrTi_2_O_7_-2M, PDF#84-0163) were also observed. This is because low CTZ content is not conducive to the crystallization of zirconolite. When the CTZ content increases, intense characteristic peaks belonging to zirconolite and powellite are observed in the CTZ-30, CTZ-40 and CTZ-50 samples. It is interesting to note that no neodymium aluminum perovskite crystals were detected in these glass-ceramics. This may be because the increase in CTZ content leads to an increase in, and during the crystallization of zirconolite, Nd^3+^ cations entered its lattice, resulting in the disappearance of neodymium aluminum perovskite in the samples. For the CTZ-30 and CTZ-40 samples, only two target crystalline phases for zirconolite and powellite were observed. However, the intense characteristic peaks belonging to perovskite (CaTiO_3_, PDF#81-0562) were detected in the CTZ-50 sample. This is because excessive TiO_2_ leads to the formation of perovskite, which is similar to what Wu reported [36]. These results are consistent with those of DTA. In conclusion, the CTZ-30 and CTZ-40 samples meet the experimental design based on the purpose of preparing multi-phase glass-ceramics containing zirconolite and powellite.

In order to study the structural changes of zirconolite crystals, Rietveld refinement of CTZ-30, CTZ-40 and CTZ-50 was performed using the JCPDS database [37,38,39]. Because three phases, zirconolite, powellite and perovskite, are detected in the XRD, multiple crystals are calculated simultaneously in the Rietveld refinement. The Rietveld refinement results are reported. As presented in Figure 4, the errors R_wp_ of CTZ-30, CTZ-40 and CTZ-50 between Y_obs_ and Y_calc_ are 7.43%, 9.20% and 7.80%, which are within the acceptable limits. Table 2 and Table 3 list the refined cell parameters of zirconolite and powellite crystals, respectively. It is interesting that the cell parameters of powellite have little changes with the increase in CTZ content, while with the increasing content of CTZ, the cell volume of zirconolite firstly increases and then decreases. The results indicate that different amounts of Nd^3+^ cations have entered the lattice of zirconolite. This is because the radius of Nd^3+^(1.109Å) is similar to that of Ca^2+^ (1.12Å) in 8-fold coordination, which enables Nd^3+^ to replace Ca^2+^ sites [40] and Ti^4+^ replaced by Al^3+^ cations to compensate charge as the following formula Ca_1-x_Nd_x_ZrTi_2-x_Al_x_O_7_ [41,42]. It is interesting that there are two obvious changes which can explain the changes in cell volume. One is the disappearance of surface crystallization. According to crystallography, one of the conditions for crystal precipitation is supersaturation of the solution. When there is a large amount of glass phase in the sample such as CTZ-30, Nd^3+^ is mainly dissolved into the glass first; when the glass phase in the sample decreases, such as CTZ-40, more Nd^3+^ cations enter its lattice in the process of zirconolite crystallization. The other is attributed to the appearance of perovskite. Nd^3+^ cations also can enter the lattice of perovskite during crystallization [43], which results in the crystals that immobilize Nd not being unique. Appendix A shows the detailed space occupying the results of zirconolite in the refined CTZ-40. According to the refinement results, zirconolite in CTZ-40 can be identified as Ca_0.78_Nd_0.22_ZrTi_1.78_Al_0.22_O_7_.

Figure 5 shows the appearance of the glass-ceramic samples. CTZ-20 shows a slight purple glass sheen with a little split-phase. Figure 5b exhibits the appearance of CTZ-30, in which the amorphous phase is surrounded by a white crystal layer. Surface crystallization can be obviously observed for both CTZ-20 and CTZ-30. This is because the insufficient content of CTZ led to the low content of the crystalline phase, and crystallization started at the surface due to the high energy at the interface. As for the CTZ-40 and CTZ-50 samples shown in Figure 5c,d, the overall color and texture are consistent. It is obvious that the homogeneity is better. It is important to avoid uneven crystallization, which can lead to the uneven distribution of elements in the glass and crystalline layers, thus affecting the entry of the simulated nuclides into the crystals. The phenomenon indicates that increasing the CTZ content not only promotes more crystals, but it also leads to a more homogeneous waste form. Combined with the XRD pattern, these results indicate that the CTZ-40 sample is the most likely of these glass-ceramic samples to be used for HLW immobilization.

Figure 6 presents SEM micrographs of the glass-ceramics and the EDS spectra of the apparent crystals in the glass matrix. As shown in the appearance results, the CTZ-20 and CTZ-30 samples show obvious phase separation (Figure 6a,b). Figure 6e,f shows a microstructure at the interface between the crystallization zone and glass phase in the CTZ-20 and CTZ-30 samples. It can be observed that many zirconolite crystal clusters are immersed in the glass matrix, and the sizes of these crystals in the clusters are about 500 nm. Similar results have been observed in zirconolite-containing borosilicate glass-ceramics [44]. This phenomenon indicated that the content of CaO, TiO_2_ and ZrO_2_ is not enough to promote zirconolite crystallization in borosilicate glass. In addition, some spherical-shaped crystals in the glass matrix for the CTZ-30 samples were observed, which corresponds to the powellite phase [45]. This is the result of MoO_4_^2-^ units separating from the glass network and then compensating the charge with Ca^2+^ cations [17,46]. It can be observed in Figure 6c,d that the CTZ-40 and CTZ-50 samples possess needle-shaped crystals with a size of about 60–90 μm in length, which is consistent with the previous work [47]. According to EDS analysis, the needle-shaped crystals correspond to zirconolite (Figure 6g). In addition, powellite crystals are mainly distributed around zirconolite crystals. However, some flaky crystals can be observed in the CTZ-50 samples. This should be related to perovskite crystallization according to EDS analysis (Figure 6h). The results obtained above indicate that the phase assemblage and microstructure of the multi-phase glass-ceramics are closely related to the content of CTZ in the glass and that the CTZ-40 sample is a candidate glass-ceramic for HLW immobilization.

In order to study the distribution of elements, elemental mapping was performed. Elemental mapping images of the CTZ-40 sample are shown in Figure 7. Ca, Ti and Zr elements were mainly concentrated in the needle-shaped zirconolite crystals, apparently. The Nd element was mainly distributed in the zirconolite crystals. The Mo element was mainly distributed in the powellite crystals. These results further indicate that Nd^3+^ cations entered into the lattice of zirconolite. Mo^6+^ cations mainly exist in the form of powellite and some of them dissolve in glass, which avoids the production of soluble molybdate.

Bulk densities and molar volumes of the obtained glass-ceramic samples are presented in Figure 8. The densities of the samples are about 2.92, 3.05, 3.24 and 3.34 g·cm^−3^. Bulk densities of the glass-ceramics show an increasing trend with increasing CTZ content. In addition, molar volumes decreased from 25.77 to 24.44 cm^3^·mol^−1^. Based on Table 1, the mean molar mass increased from 75.25 to 81.68 g·mol^−1^ with increasing CTZ content from 20 wt% to 50 wt%, which explains the increase in density.

Figure 9 presents the normalized leaching rates of Na (*LR*_Na_), Ca (*LR*_Ca_), Mo (*LR*_Mo_) and Nd (*LR*_Nd_) of the CTZ-40 samples. It can be observed that *LR*_Na_, *LR*_Ca_, *LR*_Mo_ and *LR*_Nd_ show a decreasing tendency with the increasing leaching time and tend to hold steady after 14 days. In previous studies [48,49,50,51], Si reacted with water as Si-O-Si + H_2_O→Si-O-H + HO-Si led to a dense amorphous gel layer on the surface of the sample, as a diffusion barrier, which can be used to explain the decrease of normalized leaching rates. In addition, it can be observed in Figure 9 that for CTZ-40 samples, *LR*_Na_ and *LR*_Ca_ remained almost unchanged after 14 days at around 2.2 ×10^−3^ g·m^−2^·d^−1^ and 7.2 ×10^−4^ g·m^−2^·d^−1^, which is lower than that of the zirconolite glass-ceramics reported by Wu [52]. Furthermore, *LR*_Mo_ and *LR*_Nd_ also presented a downward trend with the increasing leaching time. For the typical CTZ-40 sample, *LR*_Mo_ and *LR*_Nd_ were about 6.4 ×10^−3^ g·m^−2^·d^−1^ and 7.6 ×10^−5^ g·m^−2^·d^−1^ at 28 d, which is generally lower than that of the powellite glass-ceramics reported by Neeway [22] and the zirconolite glass-ceramics reported by Zhu [47], respectively. It can be further demonstrated that the zirconolite- and powellite-based multi-phase borosilicate glass-ceramics, especially the sample of CTZ-40 in this study, have suitable chemical durability.

## 4. Conclusions

The multi-phase borosilicate glass-ceramics containing zirconolite and powellite with different CTZ content were synthesized for RE- and Mo-rich HLW immobilization by in situ thermal treatment. All RE and Mo in 35 wt% HLW were considered, where RE were all substituted by Nd. The effects of CTZ content on crystallization, microstructure and chemical durability were investigated. The results show that only the target crystal phases of zirconolite and powellite were detected in CTZ-40. For the CTZ-20 and CTZ-30 samples, insufficient CTZ content resulted in obvious crystallization on the surface of the samples. However, when the CTZ content increased to 50 wt%, a perovskite phase appeared. These results indicate that increasing the CTZ content can promote crystallization. For the typical CTZ-40 sample, zirconolite can be identified as Ca_0.78_Nd_0.22_ZrTi_1.78_Al_0.22_O_7_, which indicates that some Nd^3+^ cations entered the lattice of zirconolite. In addition, SEM and EDS analysis further confirmed the enrichment of Nd and Mo in the two target phases of zirconolite and powellite. Furthermore, the aqueous durability of the samples with different CTZ content is appropriate. For the typical CTZ-40 sample, *LR*_Na_, *LR*_Ca_, *LR*_Mo_ and *LR*_Nd_ were 2.2 ×10^−3^ g·m^−2^·d^−1^, 7.2 ×10^−4^ g·m^−2^·d^−1^, 6.4 ×10^−3^ g·m^−2^·d^−1^ and 7.6 ×10^−5^ g·m^−2^·d^−1^ at 28 d, respectively. The above preliminary research suggests that multi-phase borosilicate glass-ceramics containing zirconolite and powellite are potential waste forms for RE- and Mo-rich HLW immobilization.

## Figures and Tables

**Figure 1 materials-14-05747-f001:**
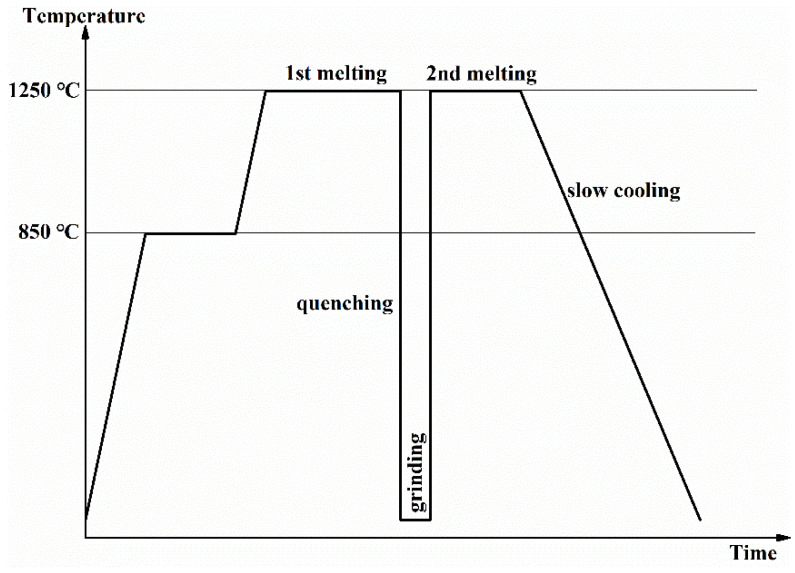
Schematic diagram of different stages of glass-ceramics preparation.

**Figure 2 materials-14-05747-f002:**
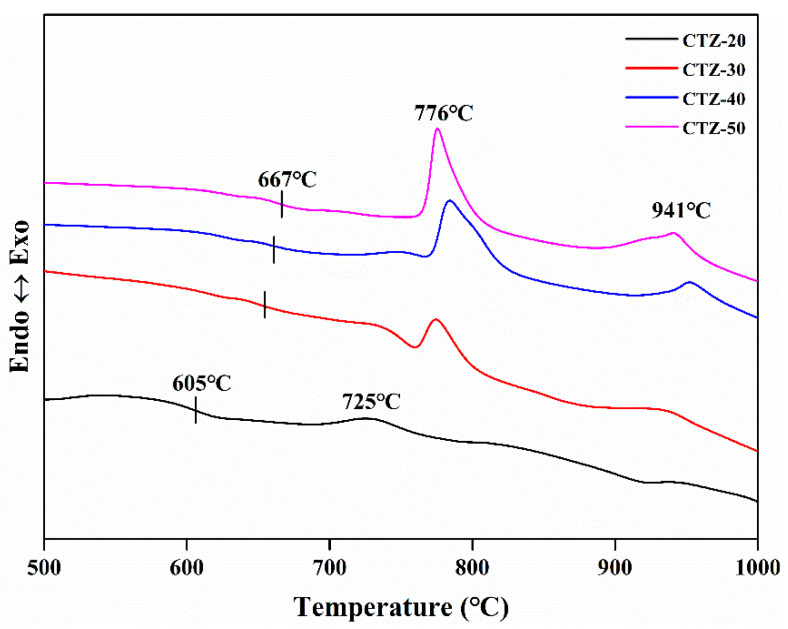
DTA curves of the glass with different CTZ content.

**Figure 3 materials-14-05747-f003:**
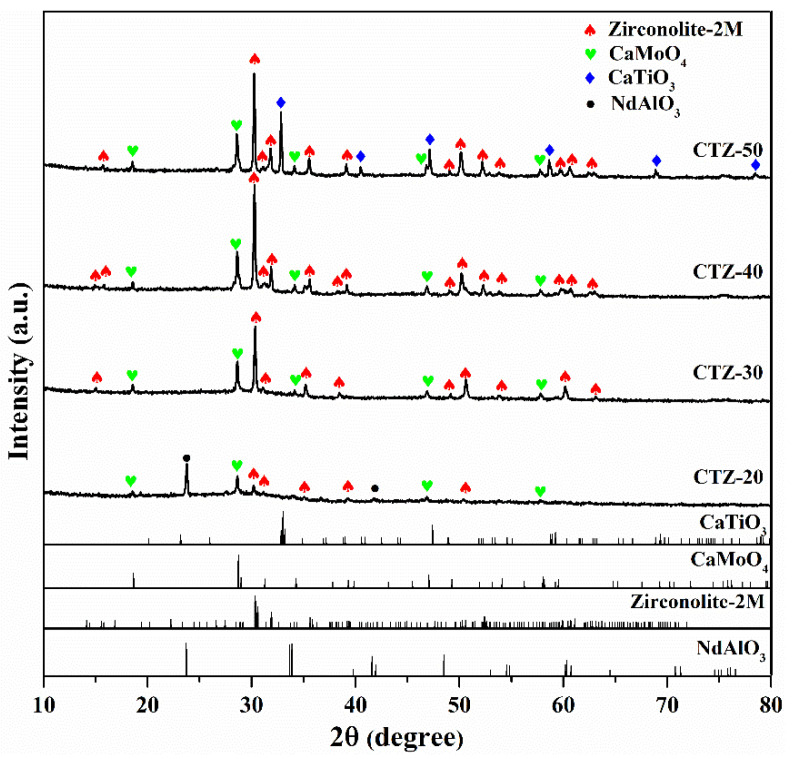
XRD patterns of glass-ceramics: CTZ-20, CTZ-30, CTZ-40 and CTZ-50.

**Figure 4 materials-14-05747-f004:**
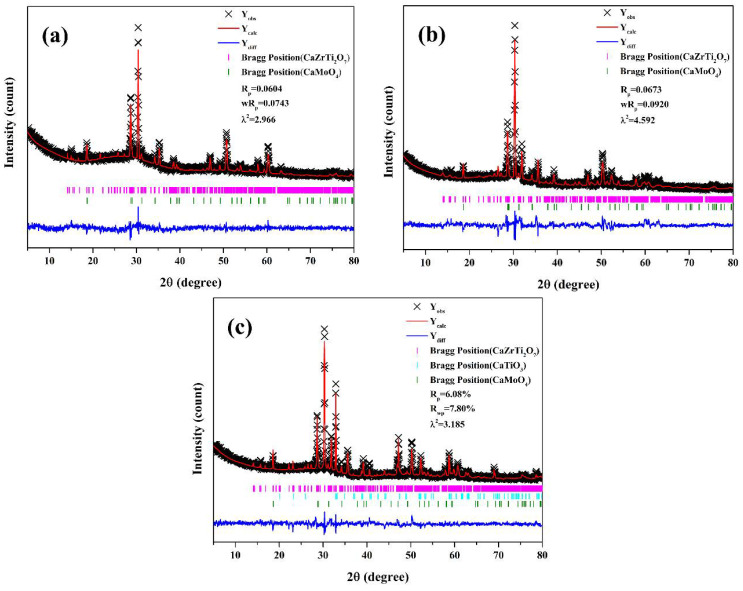
Rietveld refinement data map of the XRD of (**a**) CTZ-30, (**b**) CTZ-40 and (**c**) CTZ-50.

**Figure 5 materials-14-05747-f005:**
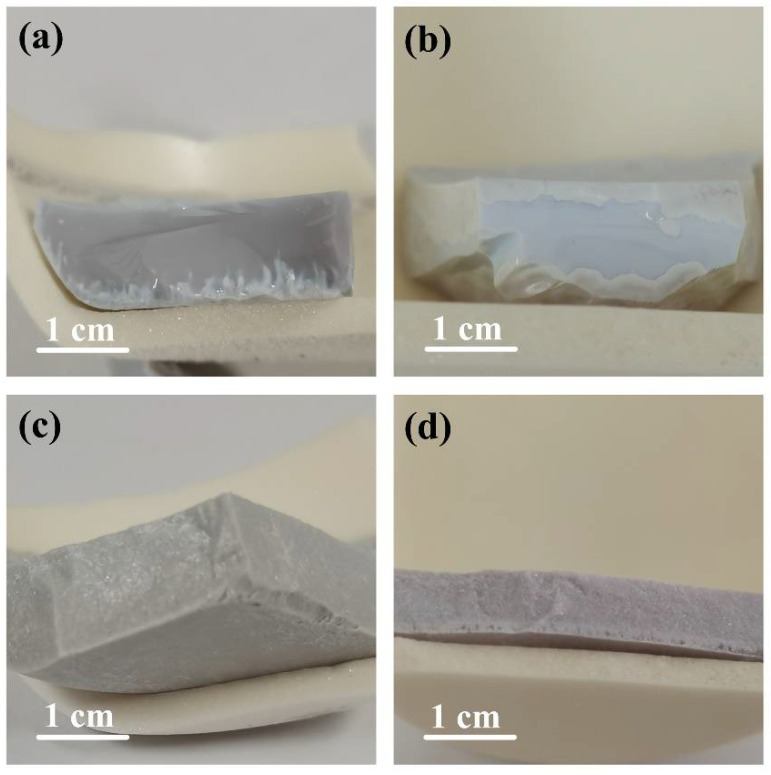
Appearance of the glass-ceramics: (**a**) CTZ-20; (**b**) CTZ-30; (**c**) CTZ-40 and (**d**) CTZ-50.

**Figure 6 materials-14-05747-f006:**
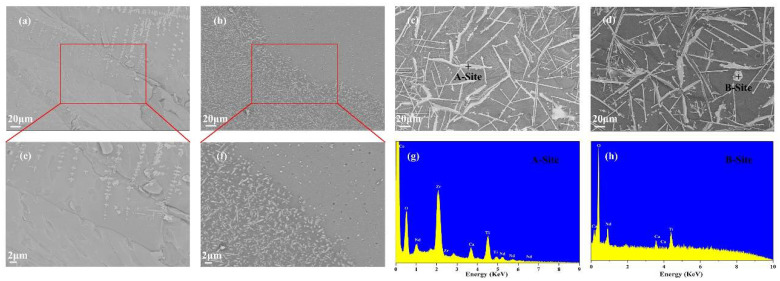
SEM images of the samples with different CTZ content: (**a**) CTZ-20, (**b**) CTZ-30, (**c**) CTZ-40, (**d**) CTZ-50; (**e**) partial enlarged view of CTZ-20, (**f**) partial enlarged view of CTZ-30; EDS spectra of CTZ-40 and CTZ-50: (**g**) A-Site, (**h**) B-Site.

**Figure 7 materials-14-05747-f007:**
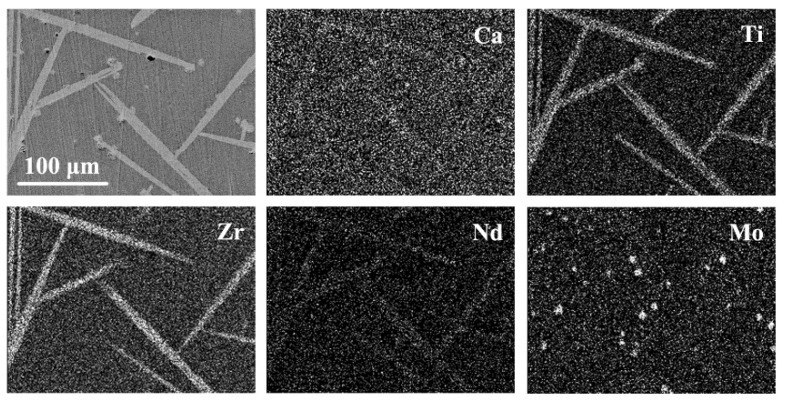
EDS maps of the typical CTZ-40 sample.

**Figure 8 materials-14-05747-f008:**
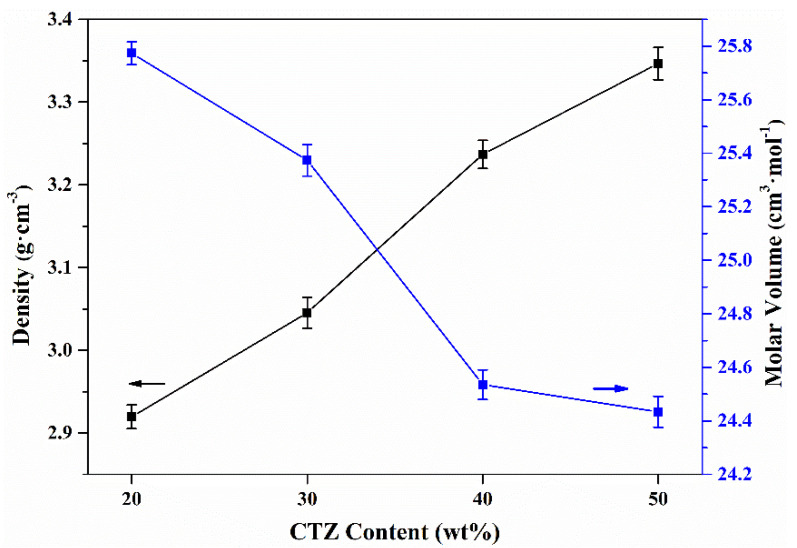
Bulk densities and molar volumes of the glass-ceramic samples.

**Figure 9 materials-14-05747-f009:**
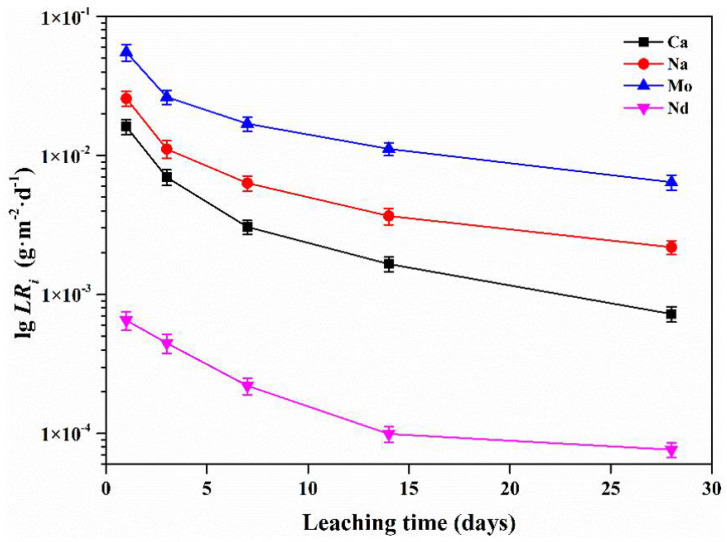
Normalized leaching rates of Na, Ca, Mo and Nd.

**Table 1 materials-14-05747-t001:** Compositions of glass matrix (mol%).

	SiO_2_	B_2_O_3_	Na_2_O	Al_2_O_3_	CaO	TiO_2_	ZrO_2_	Nd_2_O_3_	MoO_3_
CTZ-20	48.80	10.73	12.07	1.21	7.63	7.83	3.41	2.67	3.07
CTZ-30	43.85	9.64	13.16	1.08	9.04	12.07	5.25	2.75	3.15
CTZ-40	38.63	8.50	11.60	0.95	10.53	16.53	7.19	2.82	3.24
CTZ-50	33.11	7.28	9.94	0.82	12.11	21.26	9.25	2.90	3.33

**Table 2 materials-14-05747-t002:** Cell parameters obtained by refining the zirconolite phase in the obtained glass-ceramics. Numbers presented in parentheses are error standard deviations of the corresponding parameters.

Parameters	Raw Lattice	CTZ-30	CTZ-40	CTZ-50
a(Å)	12.4458	12.5298(4)	12.7909(1)	12.6149(4)
b(Å)	7.2734	7.2264(2)	7.3946(4)	7.2780(9)
c(Å)	11.3942	11.9817(4)	11.6172(1)	11.4489(3)
α(°)	90.000	90.000(0)	90.000(0)	90.000(0)
β(°)	100.533	100.095(2)	100.900(6)	100.697(2)
γ(°)	90.000	90.000(0)	90.000(0)	90.000(0)
V(Å^3^)	1014.06	1068.09(2)	1078.97(8)	1032.86(6)

**Table 3 materials-14-05747-t003:** Cell parameters obtained by refining the powellite phase in the obtained glass-ceramics. Numbers presented in parentheses are error standard deviations of the corresponding parameters.

Parameters	Raw Lattice	CTZ-30	CTZ-40	CTZ-50
a(Å)	5.226	5.241(8)	5.249(10)	5.237(1)
b(Å)	5.226	5.241(8)	5.249(10)	5.237(1)
c(Å)	11.43	11.49(2)	11.49(2)	11.47(1)
α(°)	90.0	90.0(0)	90.0(0)	90.0(0)
β(°)	90.0	90.0(0)	90.0(0)	90.0(0)
γ(°)	90.0	90.0(0)	90.0(0)	90.0(0)
V(Å^3^)	312.17	315.48(6)	316.59(9)	314.62(2)

## Data Availability

Not applicable.

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
