# Peer review of "Borosilicate Glass-Ceramics Containing Zirconolite and Powellite for RE- and Mo-Rich Nuclear Waste Immobilization"

_materials, 2021, doi:10.3390/ma14195747_

Round 1

Reviewer 1 Report

Remarks to the Authors:

  1. Title should not contain any abbreviations without their explanation (see CTZ in the title, or HLW in the capture of Table 2).
  1. Sentences marked with highlighter (see the uploaded, reviewed manuscript) should be rephrased as their English is not appropriate.
  2. Chapter 2, Paragraph 2: It seems to me, the following sentences belong together. “However, it is difficult to increase the solubility of HLW in glass, due to the limited solubility of MoO3 (~ 2.5 wt%) [8-16]. Because Mo6+ cations with high field strength may separate from glass network and combine with alkali and alkaline earth cations to form molybdate.
  1. Chapter 2, Paragraph 4: The formula of zirconolite should be given.
  2. Informal phrases (e.g. a lot of researches) should be avoided.
  3. Chapter 2, Paragraph 5: Authors write “…both of them perform great chemical durability. Excellent glass-ceramics waste forms should have excellent microstructures and uniformity, which are strongly related to composition and heat treatment.”

Instead of using informal phrases as “great chemical durability” and “excellent microstructures”, values or ranges should be added in these cases to define what "great" or "excellent" stands for. Moreover, "excellent" is repeated within the same sentence. To avoid redundancy, please, use different adjectives.

  1. Chapter 2, Paragraph 3: The formula of zirconolite should be given.
  2. When “glass-ceramic” is an adjective – showing the type of the material –, it should be in the singular.
  3. The quality of some figures (Fig 1, 2, 3, 4, 7, 8) is inappropriate.
  4. In DTA analysis, peaks correspond to a process, not to a compound. E.g. “The peak at around 940 ℃ is corresponding to powellite crystalline phase”. In my opinion, the peak corresponds to the formation/crystallization of powellite phase, not to the phase itself. The same finding should be applied to the other peaks.
  5. Using respectively is inadequate in the following sentences because it is used to relate at least two items previously mentioned in the text, meaning “separately” or “in the order given”.

Chapter 3, Paragraph 4: “It is interesting that there are two obvious changes which can explain the changes of cell volume respectively.”

Chapter 3, Paragraph 7: “The densities of the samples are about 2.92, 3.05, 3.24 and 3.34 g·cm-3, respectively.”

  1. There are some typos in the manuscript: OXford Instruments Analytical Limited, chemical formular (in chapter Conclusions), CRediT authorship contribution

  1.  The word “parameters” should be capitalised in Table 3 and 4.
  2. Chapter 3, Paragraph 5: Authors write “the size of the zirconolite crystals … which is similar to Zhu [42] and “…perovskite, whose shape is similar to Wu [41].” The size and shape of the crystals are not similar to the referred researchers, but to their results.
  3. Fig 7, X-axis: Title should be changed to “CTZ content” instead of “Contents of CTZ”

Reviewer 2 Report

The authors report glass-ceramics containing CaO, TiO2, and ZrO2.

First of all, the reader can not understand the “CTZ”.

As the authors know, the roles of CaO, TiO2 and ZrO2 in oxide glasses are different. What is the most important component among them?

Since precipitated crystalline phases in Fig. 3 are mixture of several phases, we can not understand that the main purpose (or main crystalline phase).

The relationship between Table 1 and Table 2 is also not clear.  If the authors focus on the HLW, the compositions should be designed using mol%, not wt%. Such viewpoint can be seen in Table 1, in which no systematic change in number of cations is observed.

Although SEM-EDS maps are presented, no one can understand the important point from the images. All glass-ceramics contain several precipitated phases as shown in Figs. 2 and 3. However, observed areas of SEM and EDS are not clear from the text. Therefore, it can be said that there is not direct connections among these experimental datasets.

If the authors want to claim the leaching rate of these glass-ceramics, the design of this study should be totally revised.

The contents of this paper are far from being accepted for publication in this journal.

Round 2

Reviewer 2 Report

The authors have revised the manuscript. Although the quality has been improved, the revision is needed. 

At first, definition of glass and glass-ceramics are obscure. Generally, glass-ceramics are prepared by the post heat-treatment of  precursor glass. If the heat-treatment was performed, the conditions, such as temperature, duration and atmosphere, should be shown in the text. The XRD patterns shown in Fig. 2 are not enough. All diffraction peaks shown in Fig .2 should be assigned. We can observe small diffractions in CTZ-20, -30, and -40! 

Add scale bar in Fig.4.
Scale bar in Fig. 6 is too small to be observed. 
Add the error bars in Fig. 8.

The word "glass transformation temperature" is wrong.
